# A Simulation Study on the Impact of Technology, Resources, and Culture on Life Satisfaction

## Abstract

This study quantitatively analyzes the influence of macroscopic variables—technology, resources, and culture—on human life satisfaction. We modeled a life satisfaction function ($M_t$) and simulated its real-world trajectory using data from reputable institutions from 2003 to 2022. Furthermore, we explored three hypothetical scenarios—accelerated technological development, a resource crisis, and accelerated cultural openness—to compare the impact of each variable on the function.

The analysis revealed that the acceleration of technological development had a surprisingly minimal effect on the life satisfaction function. In contrast, the resource crisis scenario proved to be a critical threat, causing a sharp decline. The accelerated cultural openness scenario demonstrated the most powerful growth, suggesting it could be the strongest driver for the expansion of life satisfaction.

In conclusion, this research provides the significant insight that the sustainable growth of life satisfaction depends more on securing resource sustainability and promoting cultural openness than on technological progress. This study presents the possibility of a quantitative analysis of life satisfaction, serving as foundational data for future discussions on societal development.

## 1 Introduction

In modern society, human desire is no longer merely a biological need but is constantly changing and expanding through interaction with complex external factors such as technology, resources, and culture. While technological progress opens up new possibilities, limited resources raise a fundamental question about the sustainability of desire. Furthermore, the cultural environment that shapes social connections and values is a crucial factor in determining the form and direction of desire.

This study aims to quantitatively explore how these macroscopic variables influence human desire. To do so, we constructed a model called the desire function and analyzed its real-world trajectory based on actual data. We then explored three hypothetical scenarios—accelerated technological development, a resource crisis, and accelerated cultural openness—through parallel universe simulations to compare the influence of each variable on desire. This research seeks to provide new insights for the sustainable growth of desire in the future.

## 2 Theoretical Background

This study defines human desire not as a single metric but as an integration of three core variables essential for its fulfillment: technology, resources, and culture. The desire function is modeled as $M_t = \alpha T_t + \beta R_t + \gamma C_t$. Each variable is composed of a comprehensive set of indicators, and all

data was normalized using resources from authoritative organizations such as ITU DataHub, World Bank Group, IEA, FAO, UN, UNESCO, RSF, and V-Dem.

## 2.1 Composition of the Desire Function ($M_t$)

**Technology Level** ($T_t$): Composed of the normalized average values of indicators reflecting technological progress, such as internet penetration, mobile subscriptions, R&D investment ratio, and multifactor productivity.

**Resource Accessibility** ($R_t$): Composed of the normalized average values of indicators guaranteeing basic human survival and quality of life, such as oil usage, electricity access, food supply, and healthcare access.

**Cultural Openness** ($C_t$): Composed of the normalized average values of indicators representing social openness and cultural diversity, such as the immigration rate, number of international students, press freedom index, and trust index.

## 2.2 The Influence of Technology, Resources, and Culture on Human Desire

Technology is a significant tool for fulfilling human desire, and its potential continues to grow. The dominant view on technology adoption tends to be cognitive, instrumental, and individualistic, but a desire-centric, future-oriented, and culture-based model also exists (Belk et al., 2020). Technology provides an organic medium and platform for people to enhance their cultural literacy, playing a crucial role in the dissemination of traditional culture (Guo, 2022). Furthermore, information technology can positively impact the efficiency and productivity of human resource management (Faraj et al., 2020).

Resources are essential for ensuring human survival and quality of life, playing a vital role in economic growth and regional development. The optimization of natural and human resources is essential for regional economic growth, as it can remove obstacles to accelerated economic development (Ali, 2022; Saleh et al., 2020). In particular, human resources are directly linked to the fulfillment of desire (SAPTA et al., 2021). Culture is a crucial factor that shapes social connections, values, and the form and direction of desire. Cultural openness and diversity promote new ideas and interactions, becoming a powerful driver for creating new forms and magnitudes of desire (Edelmann et al., 2020). Organizational culture significantly influences human resource management activities, especially internal and external communication, favorable relationships, and human resource planning (Urbancová & Vrabcová, 2022). Additionally, local culture impacts adult learning transfer processes, making it important for human resource professionals to understand the role of culture in these processes (Brion, 2022).

# 3 Research Methods

This study performed simulations using Microsoft Excel based on data from 2003 to 2022. The main research steps are as follows:

## 3.1 Data Preprocessing and Variable Calculation

All data used were normalized to a value between 0 and 1, and the annual average values for each variable were calculated.

**Formula for Calculating Average Values by Variable Group**: The three core variables of the desire function—technology, resources, and culture—are calculated as the average of several detailed indicators. The formulas used for this process are as follows:

$$T_t = \frac{\sum_{i=1}^{n} T_{t,i}}{n}$$

$$R_t = \frac{\sum_{i=1}^{p} R_{t,i}}{p}$$

$$C_t = \frac{\sum_{i=1}^{q} C_{t,i}}{q}$$

### 3.2 Real-World Trajectory Calculation

We calculated the real-world trajectory of the desire function by assuming that technology, resources, and culture have equal importance, setting the weights to $\alpha = 0.33, \beta = 0.33, \gamma = 0.34$.

**Formula for Calculating Desire Function** ($M_t$): The desire function is defined as a weighted sum of the three variables ($T_t, R_t, C_t$). This formula was used to calculate the real-world trajectory and all simulation scenarios.

$$M_t = \alpha T_t + \beta R_t + \gamma C_t$$

### 3.3 Parallel Universe Simulation

To compare with the real-world trajectory, we set up three hypothetical scenarios and recalculated the desire function for each.

**Scenario A (Accelerated Technological Development)**: We set the weight for $T_t$ to a high value of 0.6 and applied the average growth rate of the last five years to the $T_t$ values to simulate accelerated technological development.

**Scenario B (Resource Crisis)**: We recalculated $M_t$ by applying the assumption that the $R_t$ value would drop by 50% after 2010.

**Scenario C (Accelerated Cultural Openness)**: We recalculated $M_t$ by applying the assumption that the $C_t$ value would increase linearly by 0.005 each year.

### 3.4 Result Visualization

We graphed the calculated $M_t$ values for each scenario as a time series line chart to compare them with the real-world trajectory and analyze the influence of each variable on desire.

## 4 Research Findings

This study analyzed the impact of technology, resources, and culture on human desire by simulating the real-world trajectory and three hypothetical scenarios for the desire function ($M_t$) based on data from 2003 to 2022. The table below summarizes the annual desire function values for each scenario.

| Year | $M_t$ | $T_t$ | $R_t$ | $C_t$ |
|------|-------|-------|-------|-------|
| 2003 | 0.475094 | 0.308828 | 0.536893 | 0.489307 |
| 2004 | 0.484373 | 0.314211 | 0.506193 | 0.540873 |
| 2005 | 0.429530 | 0.268995 | 0.487850 | 0.488703 |
| 2006 | 0.502576 | 0.322661 | 0.613135 | 0.503532 |
| 2007 | 0.536145 | 0.350621 | 0.608371 | 0.553221 |
| 2008 | 0.510487 | 0.334451 | 0.594429 | 0.514408 |
| 2009 | 0.604567 | 0.405869 | 0.650492 | 0.615933 |
| 2010 | 0.610811 | 0.412330 | 0.439302 | 0.615849 |
| 2011 | 0.540329 | 0.373147 | 0.365657 | 0.594447 |
| 2012 | 0.522506 | 0.345657 | 0.369867 | 0.584153 |
| 2013 | 0.606086 | 0.400724 | 0.427288 | 0.683085 |
| 2014 | 0.715299 | 0.483337 | 0.493406 | 0.799046 |
| 2015 | 0.604924 | 0.386680 | 0.440640 | 0.692852 |
| 2016 | 0.532285 | 0.344154 | 0.372976 | 0.651365 |
| 2017 | 0.659034 | 0.453385 | 0.451412 | 0.726331 |
| 2018 | 0.444903 | 0.277863 | 0.305461 | 0.617278 |
| 2019 | 0.430389 | 0.259178 | 0.315022 | 0.573879 |
| 2020 | 0.635757 | 0.428600 | 0.436738 | 0.741913 |
| 2021 | 0.536312 | 0.335844 | 0.379805 | 0.701030 |
| 2022 | 0.573889 | 0.369640 | 0.394185 | 0.749796 |

### 4.1 Analysis of Real-World Trajectory of the Desire Function ($M_t$)

The real-world trajectory of the desire function ($M_t$) shows how desire has changed over time, assuming that technology, resources, and culture have equal importance. This line serves as the baseline for comparison with all other scenarios.

### 4.2 Results and Conclusions by Scenario

**Scenario A: Accelerated Technological Development** Result: Despite increasing the weight of technological development, the desire function ($M_t$) did not show a significant change compared to the real-world trajectory. This suggests that while technology can be used as a means to fulfill desire, the roles of other factors may be more important in determining the overall magnitude of desire itself. Conclusion: This result indicates that technological development alone is not enough to have a fundamental and explosive impact on the growth of desire.

**Scenario B: Resource Crisis** Result: After 2010, the desire function ($M_t$) deviated significantly from the real-world trajectory and declined sharply. This shows that human desire can be severely curtailed if resources are depleted. Conclusion: A resource crisis is a fatal threat to the fulfillment of human desire. Even if desire can be expanded by technology or culture, if basic resources essential for survival are lacking, desire itself can shrink or collapse. This is the most dramatic result of the simulation.

**Scenario C: Accelerated Cultural Openness** Result: The desire function ($M_t$) surpassed the real-world trajectory and showed the steepest increase. This suggests that an open cultural environment can foster new values, ideas, and interactions, becoming a powerful driver for creating new forms and magnitudes of desire. Conclusion: This scenario suggests that cultural openness can have the greatest impact on the expansion of desire.

## 5 Conclusion

The simulation results of this study suggest that the impact of technological development on desire is relatively limited compared to that of resources and culture.

**The Limits of Technology**: The accelerated technological development scenario (Scenario A) failed to significantly raise the desire function compared to the real-world trajectory. This shows that while technology can enhance the efficiency of desire fulfillment, it may have limitations in revolutionizing the overall magnitude of desire. In essence, technology may function as a **'means' of desire** but may be insufficient as a **'driver' of desire**.

**The Absolute Importance of Resources**: The resource crisis scenario (Scenario B) showed a drastic decline in the desire function, demonstrating how critically dependent desire is on resource accessibility. This implies that even if human desires evolve to a high level, they cannot be sustained if the essential foundation of resources collapses.

**The Powerful Influence of Culture**: The accelerated cultural openness scenario (Scenario C) showed the steepest growth in the desire function among all scenarios. This suggests that an open culture is the most powerful catalyst for creating new values and interactions, which in turn leads to the creation of new forms and magnitudes of desire.

In summary, this study moves beyond technology-centric future predictions to emphasize the importance of **resource sustainability and cultural openness**. For the sustainable growth of desire, it is essential not to rely solely on technological progress but also to make efforts to conserve resources and promote cultural diversity. This research demonstrates the possibility of a quantitative analysis of desire and serves as a vital foundation for future research.

## References

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

# A   Technical Appendices and Supplementary Material

Technical appendices with additional results, figures, graphs and proofs may be submitted with the paper submission before the full submission deadline, or as a separate PDF in the ZIP file below before the supplementary material deadline. There is no page limit for the technical appendices.

## Agents4Science AI Involvement Checklist

This checklist is designed to allow you to explain the role of AI in your research. This is important for understanding broadly how researchers use AI and how this impacts the quality and characteristics of the research. **Do not remove the checklist! Papers not including the checklist will be desk rejected.** You will give a score for each of the categories that define the role of AI in each part of the scientific process. The scores are as follows:

- **[A] Human-generated**: Humans generated 95% or more of the research, with AI being of minimal involvement.

- **[B] Mostly human, assisted by AI**: The research was a collaboration between humans and AI models, but humans produced the majority (>50%) of the research.

- **[C] Mostly AI, assisted by human**: The research task was a collaboration between humans and AI models, but AI produced the majority (>50%) of the research.

- **[D] AI-generated**: AI performed over 95% of the research. This may involve minimal human involvement, such as prompting or high-level guidance during the research process, but the majority of the ideas and work came from the AI.

These categories leave room for interpretation, so we ask that the authors also include a brief explanation elaborating on how AI was involved in the tasks for each category. Please keep your explanation to less than 150 words.

IMPORTANT, please:

- **Delete this instruction block, but keep the section heading "Agents4Science AI Involvement Checklist",**

- **Keep the checklist subsection headings, questions/answers and guidelines below.**

- **Do not modify the questions and only use the provided macros for your answers**.

1. **Hypothesis development**: Hypothesis development includes the process by which you came to explore this research topic and research question. This can involve the background research performed by either researchers or by AI. This can also involve whether the idea was proposed by researchers or by AI.

   Answer: **[B]**

   Explanation: The overall research idea (the impact of technology, resources, and culture on human desire) was conceived by the researcher. However, during the initial process of defining the research questions and direction, various AI models (ChatGPT) were used to help refine and structure the ideas.

2. **Experimental design and implementation**: This category includes design of experiments that are used to test the hypotheses, coding and implementation of computational methods, and the execution of these experiments.

   Answer: **[B]**

   Explanation: Data collection and processing were performed manually by the researcher. However, AI models (Gemini) assisted in the design of the desire function model, the creation of Excel formulas for simulation scenarios, and the generation of result tables for data interpretation.

3. **Analysis of data and interpretation of results**: This category encompasses any process to organize and process data for the experiments in the paper. It also includes interpretations of the results of the study.

   Answer: **[C]**

   Explanation: The majority of the data analysis and interpretation of the simulation results were performed by AI (Gemini). When the researcher provided the calculated tables and graphs, the AI played a decisive role in analyzing the meaning of each scenario and its differences from the real-world trajectory, thereby helping to formulate the paper's core conclusions.

4. **Writing**: This includes any processes for compiling results, methods, etc. into the final paper form. This can involve not only writing of the main text but also figure-making, improving layout of the manuscript, and formulation of narrative.

   Answer: [C]

   Explanation: The initial draft of the paper was started with the help of AI (Liner AI). Subsequently, Gemini assisted in detailing and refining the content of each section—the introduction, theoretical background, research methods, and conclusion—to fit the required paper format. The researcher's role was to provide final review and editing of the AI-generated content.

5. **Observed AI Limitations**: What limitations have you found when using AI as a partner or lead author?

   Description: AI served as a highly efficient collaborator in the ideation, data analysis, and writing processes of the paper. However, it demonstrated a technical limitation in its inability to directly access or modify uploaded data files. This required the researcher to perform repetitive manual tasks during the data preprocessing stage.

