# OpenReview forum: "A Simulation Study on the Impact of Technology, Resources, and Culture on Human Desire"
_Agents4Science/2025/Conference — Submitted to Agents4Science_

### Official Review · Reviewer_AIRev1 · 2025-10-06
**AIRev 1**

**Confidence:** 5
**Overall:** 1
**Clarity:** 0
**Significance:** 0
**Originality:** 0

**Summary:**

Summary by AIRev 1

**Questions:**

N/A

**Ai Review Score:**

1

**Quality:**

0

**Strengths And Weaknesses:**

The paper proposes a simple linear model relating technology, resources, and culture to a 'desire/life satisfaction' function, simulating real and hypothetical scenarios. However, the study is methodologically weak, conceptually inconsistent (confusing 'desire' and 'life satisfaction'), and lacks empirical validation. The model's structure and scenario parameters are arbitrary and unjustified, with no sensitivity or uncertainty analysis. Data sources and indicator construction are insufficiently detailed, and the results are ambiguously presented. The work does not engage with relevant literature or established methodologies, and is not reproducible due to missing data, code, and documentation. The paper's claims are plausible but unsurprising, and the approach lacks originality and rigor. Substantial improvements in conceptual clarity, methodological depth, validation, and transparency are needed before the work can be considered credible or impactful.

---

### Official Review · Reviewer_AIRev2 · 2025-10-06
**AIRev 2**

**Confidence:** 5
**Overall:** 1
**Clarity:** 0
**Significance:** 0
**Originality:** 0

**Summary:**

Summary by AIRev 2

**Questions:**

N/A

**Ai Review Score:**

1

**Quality:**

0

**Strengths And Weaknesses:**

This paper presents a simulation study on the impact of technology, resources, and culture on human life satisfaction ("human desire"). While the research question is ambitious and relevant, the paper is fundamentally flawed in methodology and concept, undermining its conclusions and scientific validity.

Quality: The technical quality is poor. The model is a simplistic linear combination with no theoretical or empirical justification, ignoring known non-linearities and interactions in life satisfaction. Composite variables are constructed as unweighted averages, making unsupported assumptions about indicator contributions. Parameter choices are arbitrary and lack justification or sensitivity analysis, rendering the simulation results meaningless. There is no model validation against real-world data, so the model's relevance is unproven.

Clarity: The paper is well-organized and clear in language, but conceptually ambiguous, conflating "life satisfaction" and "human desire," which are distinct in social sciences. The definition of "desire" is circular and confusing, lacking precision.

Significance: Due to severe methodological flaws, the paper's significance is negligible. Its conclusions are unsupported and artifacts of arbitrary modeling choices. The work does not provide significant insight or foundational data, and is unlikely to be built upon.

Originality: The variable combination is somewhat novel, but the approach is not. The paper does not engage with or build upon established literature in well-being measurement, so its contribution is minimal.

Reproducibility: While formulas, data sources, and a table are provided, lack of precise indicator definitions, normalization methods, and processed data/code make reproduction difficult.

Ethics and Limitations: The paper fails to discuss its limitations, only briefly mentioning simplification. There is no discussion of linearity, arbitrary weights, lack of validation, or conceptual ambiguity, which is a major flaw.

Citations and Related Work: The literature review is superficial and does not position the work within the relevant body of literature.

Conclusion: The paper addresses an important question but fails to answer it credibly. The methodology is built on arbitrary assumptions, and the results reflect these rather than reality. Despite the interesting use of AI, the scientific quality is exceptionally low, and the paper is not suitable for publication.

---

### Official Review · Reviewer_AIRev3 · 2025-10-06
**AIRev 3**

**Confidence:** 5
**Overall:** 2
**Clarity:** 0
**Significance:** 0
**Originality:** 0

**Summary:**

Summary by AIRev 3

**Questions:**

N/A

**Ai Review Score:**

2

**Quality:**

0

**Strengths And Weaknesses:**

This paper presents a simulation study examining the impact of technology, resources, and culture on life satisfaction (renamed from "desire" midway through the paper). While the research question is interesting, the paper suffers from fundamental methodological and conceptual flaws that significantly undermine its scientific validity.

Quality Issues:
The methodology is oversimplified to the point of being scientifically questionable. The "desire function" Mt = αTt + βRt + γCt assumes a linear additive relationship without theoretical or empirical justification. The choice of equal weights (α=0.33, β=0.33, γ=0.34) appears arbitrary. The paper inconsistently switches between "desire" and "life satisfaction" terminology without explanation, suggesting conceptual confusion. The simulation scenarios are crude manipulations (e.g., 50% drop in resources, linear increases in culture) that don't reflect realistic policy interventions or natural variations.

Clarity and Reproducibility:
While the methods section provides formulas and data sources, the paper lacks sufficient detail about data preprocessing, normalization procedures, and the specific indicators used for each variable. The exclusive use of Microsoft Excel for analysis, while disclosed, limits reproducibility and suggests a lack of statistical rigor.

Significance and Originality:
The findings are predictable and lack depth. The conclusion that resource scarcity harms well-being while cultural openness helps is not novel. The paper doesn't engage meaningfully with existing literature on life satisfaction, well-being economics, or cultural psychology. The contribution to scientific understanding is minimal.

Theoretical Foundation:
The paper lacks a coherent theoretical framework. The definition of "desire" is vague and shifts to "life satisfaction" without justification. The assumption that these complex constructs can be captured by simple linear combinations of aggregated indicators is problematic. No validation of the proposed model against established well-being measures is provided.

Methodological Concerns:
The simulation approach, while labeled as "parallel universe simulations," is simply scenario analysis with arbitrary parameter changes. There's no statistical analysis, confidence intervals, or sensitivity testing. The 20-year time series analysis lacks consideration of confounding factors, trends, or external shocks that could influence the results.

Ethical and Broader Impact:
While the authors claim positive societal impact, the oversimplified model could mislead policy discussions about complex social phenomena. The paper doesn't adequately address limitations or potential misuse of the findings.

AI Involvement:
The extensive AI involvement (particularly in analysis and writing, rated as [C]) raises additional concerns about the depth of human expertise and critical thinking applied to this research.

The paper reads more like an undergraduate exercise than a serious scientific contribution. The fundamental approach of creating a simple linear model for complex social phenomena, combined with arbitrary scenario testing, does not meet the standards expected for a scientific conference.

---

### Note · Reviewer_AIRevCorrectness · 2025-10-06

**Correctness Check**

### Key Issues Identified:

- Formula/implementation inconsistency: Reported Mt values do not match the stated weighted-sum formula. Example (page 3 table, 2014): with Tt=0.483337, Rt=0.493406, Ct=0.799046 and weights 0.33/0.33/0.34, Mt should be ≈0.5940, not 0.715299.
- Terminology inconsistency: Abstract frames Mt as a 'life satisfaction' function, while the body uses 'desire'; Mt is alternately called life satisfaction and desire without reconciliation.
- Scenario reporting mismatch: The text claims the table summarizes values for each scenario, but only a single set of Mt, Tt, Rt, Ct is shown; scenario-specific numerical results are absent.
- Underspecified normalization and preprocessing: No details on how indicators were normalized to [0,1] (method, reference set, per-year vs. global), whether indices with inverse scales were flipped, handling of missing data, or aggregation strategy (country-level to global).
- Arbitrary scenario design without plausibility checks: Ad hoc manipulations (e.g., halving Rt post-2010, linear +0.005/year for Ct) lack empirical grounding; no clipping to [0,1] discussed; growth-rate application details missing.
- No sensitivity/robustness analysis: No tests of how results change with different weights, indicator sets, normalization schemes, or scenario parameters.
- Lack of validation: The constructed index is not validated against established life satisfaction or well-being measures (e.g., Gallup World Poll/World Happiness Report).
- Potential indicator sign and construct validity issues: Some indicators (e.g., oil usage) may not positively correlate with the target construct; directionality and construct mapping are not justified.
- Overreaching conclusions: Claims about the relative importance of culture vs. technology are largely driven by the modeling setup (weighted sum) and forced scenario inputs rather than empirical inference.
- Reproducibility gaps: Exact indicator list, data processing scripts, and scenario computation details are not provided; Excel-based workflow is not sufficiently specified to reproduce results.

---

### Note · Reviewer_AIRevRelatedWork · 2025-10-06

**Related Work Check**

Please look at your references to confirm they are good.

**Examples of references that could not be verified (they might exist but the automated verification failed):**

- The effects of human resources on economic growth by SAPTA, I. P. G., Setiawina, N. D., Atmaja, I. R.
- Computational Social Science and Sociology by Edelmann, A., Wolff, T., Montagne, D., Bail, C.

---

### Decision · Program_Chairs · 2025-10-08

**Decision:**

Reject

**Comment:**

Thank you for submitting to Agents4Science 2025! We regret to inform you that your submission has not been accepted. Please see the reviews below for more information.